# Single-Use Flexible Bronchoscopy: Advances in Technology and Applications

**DOI:** 10.3390/diagnostics16010150

**Published:** 2026-01-02

**Authors:** Siti Amanina Azman, Marcus Peter Kennedy

**Affiliations:** Department of Respiratory Medicine, Cork University Hospital, T12 DC4A Cork, Ireland

**Keywords:** bronchoscopy, single-use flexible bronchoscopy, disposable bronchoscopy, bronchoalveolar lavage, environmental impact

## Abstract

With advances in scope and imaging technology, the use of single-use flexible bronchoscopy (SUFB) has broadened beyond intensive care units and operating rooms to bronchoscopy units, with an expanding body of literature suggesting adequate and comparable procedure outcomes, including airway inspection, bronchoalveolar lavage, endobronchial brushing and endobronchial biopsy, in comparison to standard reusable flexible bronchoscopy (RFB). Advantages such as mobility, ease of use and lack of requirement for cleaning staff during the COVID-19 pandemic led to a global increase in usage, with many companies developing SUFB as part of their bronchoscopy portfolio. In parallel, there has been more attention and initiatives to minimise the risk of infection transmission related to bronchoscopy. RFB requires maximum adherence to manufacturer-recommended cleaning protocols. However, evidence of transmissible organisms after cleaning is reported in healthcare settings of all types. After initial benchtop, retrospective and single-arm studies, comparative bronchoscopy studies are identifying that SUFB are as versatile and non-inferior to RFB. However, cost-effectiveness and sustainability factors have to be included in deciding the use of SUFB in routine practice.

## 1. Introduction

Since the initial use of an oesophagoscope by Gustav Killian to retrieve a pork bone from a Schwarzwälder’s airway in 1876 [1], bronchoscopy technology has progressed. Since 1966, flexible bronchoscopy has revolutionised the diagnosis and treatment in the field of respiratory medicine, especially in respiratory infections and malignancy. Endobronchial ultrasound and navigational bronchoscopy have advanced beyond the proximal airways. However, a common issue in all flexible bronchoscopic technologies is that they are not typically sterilised prior to use due to equipment damage that would ensue. Naturally, the possibility of transmission of COVID-19 (coronavirus of 2019) was feared [2,3,4]. In parallel, throughout all healthcare, there was a global recommendation to shift towards single-use devices where feasible to reduce the risk of [5].

A single-use flexible bronchoscope (SUFB) is a disposable medical device used for viewing and treating the airways, which is discarded after a single use on a patient. Unlike reusable flexible bronchoscopes (RFB), these devices come sterile and ready-to-use, which reduces the risk of cross-infection and eliminates the need for cleaning, repair, and storage of the scope. SUFB uses scopes which connect typically to a portable monitor, allowing sterile procedures not limited by poorly mobile, burdensome equipment. In this review, we will briefly review the steps in cleaning reusable scopes and data regarding infection, give an up-to-date analysis of SUFB scope types, benchtop and clinical data comparing types and performance to RFB, and finally, information regarding cost and environmental impact. This article is an invited narrative review. In parallel to their clinical experience and previous publications regarding the subject, the authors have performed an up-to-date review of multiple databases and other sources to identify all eligible studies with no language or date restrictions to avoid bias.

## 2. Reusable Flexible Bronchoscopes and Risk of Infection

RFBs are semi-critical devices (Spaulding classification) that contact mucous membranes, requiring high-level disinfection (HLD) after each use. Full sterilisation is not used, as modern bronchoscopes cannot withstand autoclaving or repeated chemical sterilisation [6].

HLD involves thorough manual cleaning plus a disinfecting agent, which, for instance, achieves a six-log reduction of mycobacteria. Despite this, reusable bronchoscopes are linked to more infection outbreaks than any other medical device, largely due to reprocessing errors and underreporting [7,8].

Reprocessing is complex, time-consuming, and costly (pre-cleaning, leak testing, manual cleaning, HLD, rinsing, drying, and storage) (Figure 1A). Non-compliance with protocols is the main cause of cross-contamination [8].

Delayed cleaning allows biofilm formation inside narrow channels and surface defects. These biofilms are hard to remove and often harbour multi-drug-resistant “superbugs,” leading to documented outbreaks [7,8].

A recent high-impact study of 24 clinically used flexible bronchoscopes across three hospitals found that every scope retained residual contamination after manual cleaning. Even after full reprocessing, 58% still cultured pathogens, including mould, *E. coli/Shigella*, and *Stenotrophomonas maltophilia*. Visual inspection revealed damaged scopes with scratches, broken parts, residual fluid, greasy or dark residue, and filamentous debris in channels. Reprocessing procedures failed to meet standards at two of the three sites, demonstrating persistent contamination and significant deficiencies in current cleaning and disinfection practices [9].

ETO2 (ethylene oxide sterilisation) is a relatively new, FDA-cleared, HLD system specifically designed for flexible bronchoscopes (sometimes marketed as “the 20-min bronchoscope reprocessing solution”). In time, it may prove a solution to cross contamination; however, at present, it lacks data, and global uptake is limited by cost.

## 3. What Are Single-Use Flexible Bronchoscopes?

An SUFB is a sterile, ready-to-use bronchoscope that is used for one patient only and then thrown away. A complete SUFB set is delivered sterile in one sealed package and contains only the scope itself and suction connections (Figure 1B,C). The scope is connected to a dedicated reusable cleanable processor/monitor unit (Figure 1D), which can also be connected to standard bronchoscopy monitor interfaces. The monitors are portable, allowing ease of use and mobility. A transition from the ICU to the endoscopy suite required scope design to progress to devices similar to RFB scopes, and both COVID-19 and the awareness of scope-related infection led to multiple companies releasing SUFB technology. The number of companies with SUFBs is increasing, with a recent publication identifying 15 commercially available devices [10].

Table 1 summarises many scenarios where SUFB are advantageous over RFB. They include training and research, and other scenarios where scope trauma or damage is vastly less expensive than RFB (the requirement for decommissioning an RFB after use in a patient suspected of prion disease and potential damage related to flexible bronchoscopy through a rigid bronchoscope and with electrocautery).

## 4. Initial Intensive Care Unit Use

Ambu© introduced the world’s first single-use flexible bronchoscope in 2009, initially for intubation [11]. Since then, disposable scopes have been increasingly used for percutaneous tracheostomy, bronchoalveolar lavage, hemoptysis management, and other procedures. Most published evidence remains limited to case series and retrospective studies, with few prospective, rigorously controlled trials [12,13,14,15,16].

Early Ambu© aScope™ generations [1,2,3,4] had a distinctly different handle design compared with traditional reusable bronchoscopes. Subsequent models, including the Ambu© aScope 5™ and offerings from other manufacturers, have progressively improved ergonomics, suction performance, flexibility, and overall procedural capabilities (Figure 1C).

## 5. Benchtop Comparisons

With this increase in SUFB types available in the last 5 years, several benchtop studies have compared single-use flexible bronchoscopes (SUFBs) [17,18]. One study from our institution (2021–2022) evaluated all CE-marked models available at the time with a 2.8–3.0 mm working channel: Ambu^®^ aScope4™ Large, Boston Scientific^®^ EXALT™ Model B Large, The Surgical Company Broncoflex© Vortex, Vathin^®^ H Steriscope™ Large, and Pentax^®^ Medical ONE Pulmo™ (Figure 1B). Scope channel diameters were 2.8–3 mm [19]. Testing included measurements of size/weight, handle ergonomics, flexion/extension angles (with and without instruments in the channel), and suction performance using a pseudo-mucus liquid. Physician users (ICU and pulmonary specialists) also tested the scopes on a low-fidelity simulator.

Results showed differences in preferred handle design (varying by gender and hand size) and in maintaining angulation with instruments in place. In simulation (excluding suction and image quality), the Broncoflex© Vortex was rated highest. However, the Boston Scientific^®^ EXALT™ Model B dramatically outperformed all other single-use scopes—and even a reusable bronchoscope with a 3.2 mm channel—in suction capacity. A sacrifice, however, is that it required the maximal thumb force but had the least reduction of tip movement with forceps. The Vathin SUFB had the biggest range of tip movement from flexion to extension, with and without forceps. Although there was no significant difference in preference in the overall group, females and those with smaller hand sizes preferred the Pentax^®^ SUFB and males preferred the Broncoflex© SUFB. Other studies have confirmed adequate BAL recovery with single-use scopes compared with historical reusable controls [20]. A recent study compared five SUFBs’ BAL capabilities on a low-fidelity simulator. All were satisfactory; however, the Ambu© aScope 5™ with integrated sampling system was rated the highest [21]. These benchtop differences are expected to influence real-world clinical performance. Other factors that can influence the selection of SUFB left/right rotation capability for the insertion cord (Ambu© aScope 5™ and Vathin^®^ H Steriscope™) and the license for electrocautery (Ambu© aScope 5™). In our experience, the superior suction capabilities of the Boston Scientific^®^ EXALT™ Model B make it our scope of choice for patients with hemoptysis, thick secretions (including cystic fibrosis) and foreign bodies and for debulking endobronchial tumours [22].

## 6. Single-Use Flexible Bronchoscopy in the Bronchoscopy Suite

Publications regarding the performance of SUFBs in the bronchoscopy suite have lagged behind their increasing commercialisability (Table 2). It could be argued that, in a similar fashion to the release of new generations or sizes of RFB scopes, pilot clinical studies are not required. Publications are, in general, limited to single-arm prospective [22,23,24,25,26] trials. Of note, three of these publications are from our institution with similar methodology [22,24,25]. To assess performance, outcomes included user satisfaction and the requirement to transition to RFB. Subsequent small unblinded randomised studies followed [27,28]. Table 2 summarises the currently published data. In total, six trials have included 670 patients using five different scope types (Ambu^®^ aScope4™ and 5™, Boston Scientific^®^ EXALT™, Broncoflex© Vortex, and Vathin^®^ H Steriscope™) It is important to state that these trials are limited to populations. However, 149/670 cases (22%) included diagnostic and therapeutic procedures other than BAL (endobronchial brush and biopsy, cryobiopsy, transbronchial needle aspiration and biopsy, debulking, electrocautery, argon plasma coagulation and stent placement). Figure 2 details a clinical case of the management of a malignant airway obstruction using the Ambu^®^ aScope 5™. 21/670 (3%) required conversion to SUFB. Technical limitations and complications (including scope damage, limitation in angulation and reach and image quality) were reported in 61/670 (9%). As there was no control, many of the issues with reach and angulation could also have occurred with RFB. There were no patient-related complications. In the single-arm studies (630/670, 94%), the optimal score for satisfaction (for example, 5 on a 1–5 Likert scale) ranged between 80 and 88%. In the comparator studies, SUFB was considered comparable with RFB in all cases.

To date, no major international respiratory society has published formal guidelines or position statements on the use of SUFBs.

The only professional body that has issued detailed guidance is the Respiratory Branch of the Chinese Medical Association [29]. In their 2023 expert consensus, a panel addressed nine core aspects of SUFB use: terminology and definitions, device design and construction, clinical advantages, recommended application scenarios, pre-operative preparation, sedation and anaesthesia considerations, post-procedure handling (including disposal), and training requirements.

The document includes twelve specific consensus recommendations. Among the key points are the recognition of SUFBs’ superior portability, immediate availability, and elimination of reprocessing-related infection risk. The panel particularly emphasised that the safety and effectiveness of single-use bronchoscopes depend heavily on structured, standardised training programmes for operators, underscoring that the devices are not inherently “easier” to use and that proper skill acquisition remains essential. Thus, while clinical adoption of SUFBs continues to grow worldwide, formal endorsement or practice recommendations from most national and international respiratory societies are still awaited. As stated, thus far, published clinical experience is, in general, limited to single-arm studies of bronchoscopy with lavage. Benchtop comparisons have supported similar outcomes, such as lavage volume to RFBs. In a similar fashion to the fact that companies releasing new reusable bronchoscopes with different channels or external diameter sizes tend not to proceed to comparative clinical trials to prior iterations, it could be argued that clinical trials comparing SUFB and RFB BAL adequacy are not required and indeed could not be blinded. For standard endobronchial biopsy, brush and transbronchial needle aspiration, published literature thus far has not identified any reasons why SUFB would be inferior to RFB. Regarding recommending SUFB for more advanced interventional procedures, clearly, there is not enough data at present, and most SUFB scopes are not licensed for electrocautery procedures. However, the reality is that the majority of bronchoscopy procedures involve BAL or bronchial washes for infection and inflammation.

In general, limitations in bronchoscopy (SUFB or RFB) often arise as regards scope size, suction, stiffness and flexibility. Our prior benchtop analysis identified advantages and disadvantages of five different SUFB scopes in comparison to RFB [19]. An example of a limitation related to a particular scope is scope stiffness (Boston Scientific^®^ EXALT^TM^ Model B). In our clinical trial of this scope, access to upper lobes for biopsy or TBNA with the instrument in situ was overcome by allowing the tip back into neutral position, passing the instrument to the tip of the scope and then flexing the scope tip and biopsy [22]. However, the superior suction of this scope made it the scope of preference where thick secretions or haemoptysis were expected. Other advantages and disadvantages are highlighted in benchtop and clinical publications discussed above.

## 7. Cost Comparisons

When considering cost, there are many variables to consider, including scope manufacture, scope cleaning, staff requirements, storage and associated equipment [30,31,32,33]. Table 3 summarises current data. When a healthcare system is evaluating whether to switch from RFBs to SUFBs, cost is often the first factor raised, but it cannot be assessed in isolation. It is important to state that a variable factor as regards the cost of commencing reusable bronchoscopy is whether or not a bronchoscopy service is being started from scratch or whether equipment such as processors, monitors and cleaning processors is already present. Cost comparisons typically assume that this equipment is present already and the comparison relates to switching from RFB to SUFB.

Studies consistently show that in very high-volume centres performing more than 1200–1500 bronchoscopies per year, reusable scopes appear less expensive per procedure (typically EUR 78–EUR 150) compared with single-use scopes (around EUR 220–EUR 232), largely because the substantial fixed costs of reprocessing, staff, repairs, and accreditation are distributed across many cases. In lower-volume settings with fewer than 300–350 procedures annually, the same fixed costs are spread over far fewer cases, and single-use bronchoscopes frequently become cost-neutral or even less expensive.

Beyond the direct purchase price, single-use bronchoscopes require fewer personnel for reprocessing, eliminate the need for repairs, microbial surveillance, and maintenance of certified reprocessing areas, shorten turnaround time between cases, reduce infection risk to both patients and staff, lower occupational stress and chemical exposure, ensure immediate availability for emergencies, and provide a consistently functional scope for training.

The decisive variable, however, is the risk of cross-contamination and infection associated with reusable scopes. A comprehensive systematic review and meta-analysis of sixteen studies calculated a weighted average infection or cross-contamination rate of approximately 2.8% with reusable bronchoscopes, whereas no such events have been documented with single-use devices (expected risk 0%). When the considerable costs of diagnosing and treating these infections are included, the true cost per patient for a reusable bronchoscope rises sharply—for example, from roughly GBP 249 to GBP 511 in one detailed United Kingdom analysis—while the cost of a single-use bronchoscope remains unchanged at approximately GBP 220 [30].

Thus, analyses that consider only equipment and reprocessing expenses tend to favour reusable scopes in busy centres. Once the real-world clinical and financial consequences of scope-related infections are fully accounted for, single-use flexible bronchoscopes emerge as cost-effective and markedly safer in almost every setting.

## 8. Role in Training

SUFBs offer significant potential for medical training, particularly in simulation-based learning. Unlike reusable flexible bronchoscopes (RFBs), which are costly to purchase, reprocess, and repair, SUFBs provide a more economical option for training purposes. The high operational costs and risk of damage associated with RFBs make them impractical for repeated use in educational settings.

Our institution paired SUFBs with a low-cost bio-simulator [34]. This model enables trainees to practise and refine essential skills, such as scope handling, endobronchial biopsy, and brushing, in a realistic yet cost-efficient simulated environment. It differentiated novices from experts and led to improvement in trainees’ skills on a modified validated training tool. By leveraging SUFBs and affordable simulators, this approach enhances accessibility to high-quality training while minimising financial and logistical burdens. Other advantages for training and practice include the vastly lower expense of damaging an SUFB in comparison to an RFB related to scope trauma during rigid bronchoscopy or electrocautery.

## 9. Environmental Impact

Healthcare systems account for approximately 4–5% of global greenhouse gas, with medical devices estimated to account for 21% [35]. It is imperative that bronchoscopists consider all aspects of their practice to reduce waste [36]. An important consideration when choosing between reusable and single-use flexible bronchoscopes is their environmental impact, although the evidence remains limited and somewhat conflicting.

Few studies have performed a full life-cycle assessment of single-use bronchoscopes. One such analysis compared CO_2_-equivalent emissions and resource use between an SUFB and an RFB [37]. It found that the environmental footprint is heavily influenced by the materials used in manufacturing the single-use device and, equally importantly, by the consumables (detergents, disinfectants, water, and personal protective equipment) required for reprocessing the reusable scope. Because reprocessing practices vary widely between institutions, the authors concluded that no firm, generalisable superiority of one system over the other could be established.

A 2021 waste audit involving 278 endoscopic procedures estimated that complete conversion to single-use bronchoscopes would increase overall clinical waste by as much as 40% [38]. Similarly, a systematic review published this year, which examined multiple types of endoscopes (including bronchoscopes), again reported a more favourable environmental profile for reusable instruments when proper life-cycle methodology was applied [39].

In summary, while single-use bronchoscopes clearly offer advantages in infection prevention, readiness, and training, current evidence suggests that—from a purely environmental perspective—well-managed reusable systems still tend to have a lower carbon footprint and generate less waste, provided reprocessing is efficient and high-volume. The environmental equation, therefore, adds another layer of complexity to the decision, and local circumstances (procedure volume, waste-management infrastructure, and reprocessing efficiency) remain critical in determining the most sustainable choice. It is hard to get over the fact that this is another disposable device with plastic and electronic components that we cannot recycle. Methods to make SUFB parts recyclable need to be developed, allowing effective decarbonisation strategies. Multi-stakeholder assessments between healthcare facilities, environmental experts and the health technology sector are essential to produce transparent and objective assessments.

## 10. Conclusions

Both COVID-19 and the recognition of scope-associated infection have propelled the development of SUFB comparable with RFB for use in the bronchoscopy suite. Research has lagged scope uptake; however, it is unclear what level of research is required for increased uptake in devices that are commercially available and equivalent in limited single-arm and cost-comparison studies in association with prior ICU-based research. Cost considerations include the number of scopes performed per unit per year and the rate of RFB-related infection. Many other advantages, including ease of use, portability and a lack of requirement for cleaning staff, need to be considered. Environmental impact data varies, and like most single-use devices, producers need to figure out ways to recycle components.

While clinical adoption of SUFBs continues to grow worldwide, formal endorsement or practice recommendations from national and international respiratory societies will help increase their use.

## Figures and Tables

**Figure 1 diagnostics-16-00150-f001:**
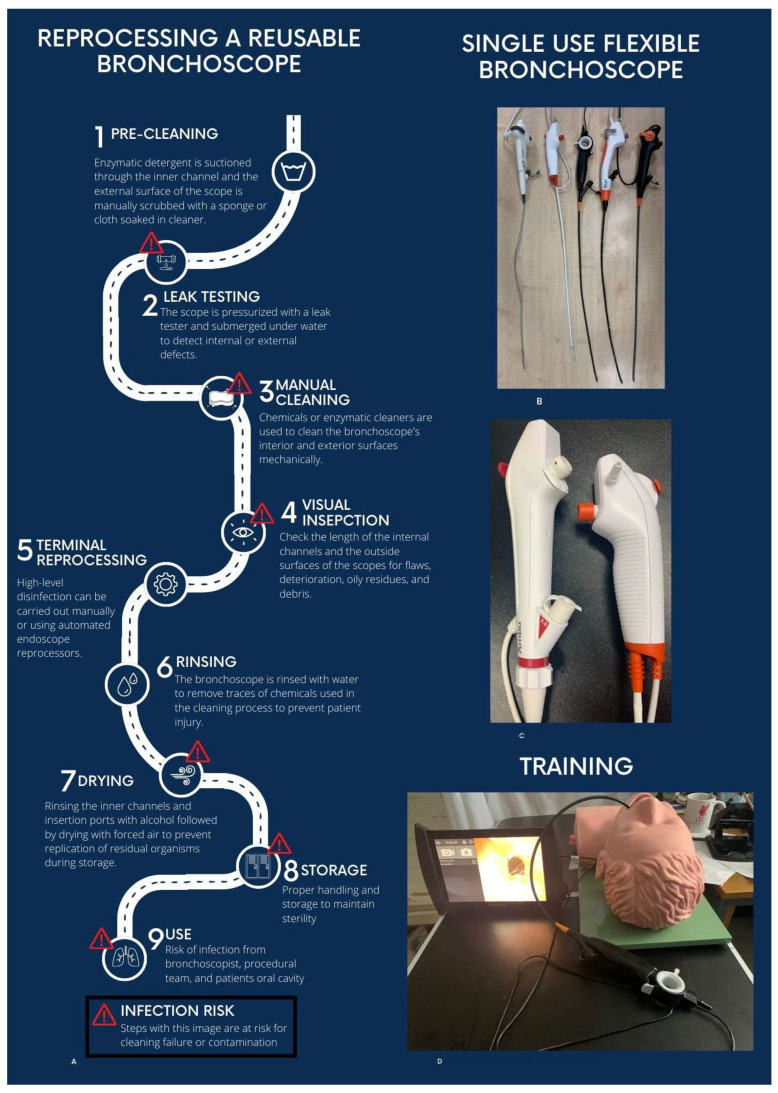
Reusable scope processing and single-use bronchoscopy. (**A**) Pathway depicting reprocessing step for a reusable bronchoscope. Steps at risk for contamination and cleaning failure are indicated by the red exclamation mark and warning triangle. (**B**) Five commercially available single-use bronchoscopes, from left to right: Pentax^®^ Medical ONE Pulmo™, Ambu^®^ aScope 4 Large (Tokyo, Japan), The Surgical Company Broncoflex© Vortex (Amersfoort, The Netherlands), Vathin^®^ H-Steriscope™ Large (Xiangtan, Hunan Province, China)and Boston Scientific^®^ EXALT™ Model B. C Ambu^®^ single-use bronchoscopes (Marlborough, MA, USA). Reprinted with permission from (**C**). On the left is the Ambu^®^ aScope™ 5 Broncho Large (Ballerup, Denmark). On the right is the Ambu^®^ aScope 4 Large. Note the progression in scope handle design with ability to rotate 120° left and right. (**D**) Axess Vision Single-Use Flexible 2.8 mm Channel Vortex with the Screeni^®^ viewing system placed in a mannequin used for bronchoscopic training. Reproduced with permission [8].

**Figure 2 diagnostics-16-00150-f002:**
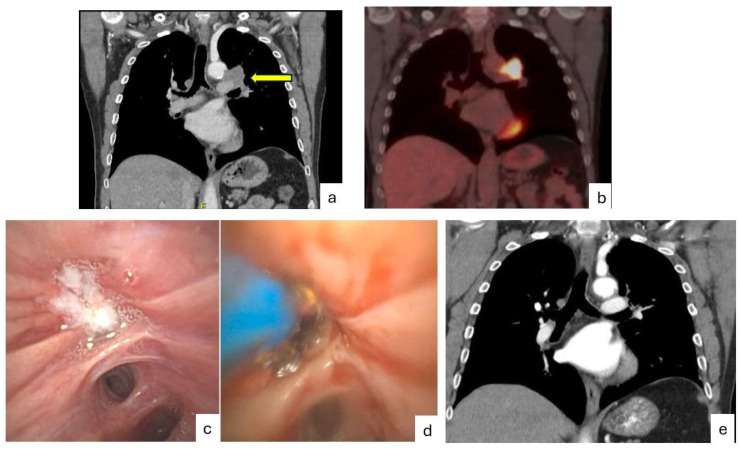
57-year-old male with non-small cell lung cancer (**a**). Sagittal CT thorax with contrast image with left upper lobe T4N0M0 indicated by yellow arrow (**b**). Sagittal fused PET-CT indicating PET-avid left upper lobe mass (**c**). Endobronchial image using the Ambu^®^ aScope™ 5 2.8 single using bronchoscope, identifying malignant airway obstruction in the anterior segment of the left upper lobe (**d**). Argon plasma coagulation of the left upper lobe tumour using the Ambu^®^ aScope™ 5 2.8 single using bronchoscope. (**e**) Sagittal CT thorax with contrast image showing response of the tumour to neoadjuvant chemotherapy and immunotherapy. The patient declined subsequent surgery.

**Table 1 diagnostics-16-00150-t001:** Clinical and other scenarios where single-use flexible bronchoscopes (SUFBs) have advantages over reusable flexible bronchoscopes (RFBs).

Ease of Mobility	Practicality	Specific Scenarios Where Reduced Risk of Cross-Infection Is Critical	Other Applications
ICU Bronchoscopy	Out-of-hours bronchoscopy	Immunocompromised patient	Bronchoscopy Training
Emergency Department/Ward Bronchoscopy	End of day list—staff are not required to stay and clean scopes	Prion Disease	Veterinary Procedures
Emergency Bronchoscopy outside Healthcare Facility	Weekend bronchoscopy where staff are not available to clean scopes		Large animal or cadaveric research
	Bronchoscope available for airway inspection with EBUS procedures		

**Table 2 diagnostics-16-00150-t002:** Single-use flexible bronchoscopy in the bronchoscopy unit: publications to date.

Author, Year, Journal	Scope	Design	Number	Procedures (Excluding BAL)	Conversion to RFB	Satisfaction	Technical Limitations
Flandes et al. Respir Res (2020) 21:320 [26]	Ambu^®^ aScope^TM^ 4	Single Arm Prospective	300	17/300 (5%) biopsy	17/300 (6.7%)	80% satisfied	17/300 (6%) 2 scopes damaged/ruptured 10 undefined 5 image
Sweeney et al. Respiration (2022) 101 (10): 931–938 [24]	The Surgical Company Broncoflex©	Single Arm Prospective	139	40/139 (29%) Biopsy, brush TBNA, APC Cryobiopsy, Stent	4 (2.8%)	83% very satisfied (5 on 1–5 Likert Scale)	22/139 (15%) image size (9) suction (8) 3 suction connector break (3) cable break (1) reach (1)
He et al. BMC Pulm Medicine (2023) 23 (202): 1–9 [27]	Vathin^®^ H-SteriScope^TM^	Controlled study SUFB vs. RB Airway obstruction	SUFB 30 vs. RB 15	15/15 (100%) biopsy	0 (0%)	Comparable with RFB	0
Tangney et al. J Thor Dis (2025); 17(1):42–50 [25]	Ambu^®^ aScope^TM^ 5	Single Arm Prospective	98	10/98 (10%): biopsy, brush TBNA, APC	0 (0%)	87% very satisfied (5 on 1–5 Likert Scale)	4/98 (4%) Photo application (2) Stiff scope (1) 1 reach (1)
Tangney et al. J Thorac Dis 2025;17(10):7585–7593 [22]	Boston Scientific^®^ EXALT™ Model	Single Arm Prospective	108	57/108 (53%) Biopsy, brush TBNA, APC Cryobiopsy, Stent	0 (0%)	88/108 82% very satisfied (5 on 1–5 Likert Scale)	18/108 (17%) difficulty passing tools (7) image (5) suction (3) angulation issue (2) stiffness (1)
Popovic et al. Pulmonology (2025) 31 (11): 2443218 [28]	Ambu^®^ aScope^TM^ 5	Controlled study SUFB vs. RB Airway obstruction	SUFB 10 vs. RB 10	10/10 (100%) Laser, electrocautery, balloon dilatation, cryoprobe, stent	0 (0%)	Comparable with RFB	0
Total			670	149/670 (22%)	21/670 (3%)		61/670 (9%)

SUFB = single-use flexible bronchoscopy. RFB = reusable flexible bronchoscopy. TBNA = transbronchial needle aspiration. APC = argon plasma coagulation. With <756 procedures a year, SUFBs may provide a more cost-neutral approach. SUFB = single-use flexible bronchoscopy.

**Table 3 diagnostics-16-00150-t003:** Cost-effective analysis including systematic reviews and meta-analysis of single-use in comparison to reusable bronchoscopes.

Author	Year of Publication	Location	Comparison	Cost/Procedure	
				SUFB	Reusable Scope
Châteauvieux et al. [31]	2018	Academic institution	1644 procedures/year	EUR 232	EUR 78
			328 procedures/year	EUR 232	EUR 232
Mouritsen et al. [30]	2020	Perioperative setting in high-throughput tertiary setting	Without 2.8% infection rate *	GBP 220	GBP 249
			With 2.8% infection rate *	GBP 220	GBP 511
Anderson et al. [33]	2022	US Academic Institutions		USD 289	USD 266
Kristensen et al. [32]	2023	Three University Hospitals and Academic Institutions	2200 procedures/year for each institution	USD 401 ^§^	USD 274

* Presuming a 2.8% risk of infection related to reusable scope contamination. ^§^ Change to SUFB would incur an additional cost of USD 129 per procedure but decrease cross-infection risk by 0.22%.

## Data Availability

No new data were created or analyzed in this study. Data sharing is not applicable to this article.

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
