# Peer review of "Single-Use Flexible Bronchoscopy: Advances in Technology and Applications"

_diagnostics, 2026, doi:10.3390/diagnostics16010150_

Round 1

Reviewer 1 Report

Comments and Suggestions for Authors

The manuscript is interesting and well structured. However, certain sections contain some redundancy (e.g. regarding infection risk and reprocessing). The methodology of the literature search is not clearly specified. A brief description of the search strategy in Material and methods section is recommended. Hier, could be used including inclusion and exclusion criteria, time frame, and search keywords.

In addition, a slight moderation of the text is suggested in order to avoid the impression that all centers experience uniformly high failure rates. 

Finally, a limitations section should be added. In particular, the limitations of single-use flexible bronchoscopes should be explicitly addressed, including their restricted applicability in interventional procedures.

Reviewer 2 Report

Comments and Suggestions for Authors

This manuscript provides a clear and valuable overview demonstrating that single-use flexible bronchoscopes (SUFBs) have now reached a level of performance that is sufficient for routine clinical practice. The authors successfully highlight their particular strengths, including their advantages in situations requiring substantial suction capability and in cases where infection risk must be minimized. The discussion of cost considerations is also appreciated, as SUFBs may offer meaningful benefits in settings where the financial burden of maintaining a reusable bronchoscopy system is substantial.

In addition, it may be worth noting that the extremely high capital costs associated with the initial installation of reusable endoscopy systems remain a major barrier for many institutions. From this perspective, SUFBs have the potential to serve as a practical solution, helping to lower the threshold for establishing bronchoscopy services. They may also contribute to reducing market dependence on a small number of established manufacturers and, in the long term, could encourage broader participation and innovation from new entrants in the endoscopic device industry. Including a brief comment on these aspects would further strengthen the manuscript by highlighting the broader implications of SUFB adoption.

That said, the manuscript would be further enhanced by providing more concrete examples of the technical limitations observed with SUFBs and clarifying the specific clinical situations in which these limitations become relevant. Although issues such as reduced angulation, reach, image quality, or instrument passage are mentioned, readers may still find it challenging to understand the scenarios in which these limitations are most likely to affect clinical practice. A more detailed description of representative cases, along with potential strategies to overcome or mitigate these constraints, would offer valuable guidance for clinicians and help shape the future direction of SUFB technology.

Overall, this is an informative and timely review. Further elaboration on the context of technical limitations and the broader economic and industry implications of SUFB adoption would significantly enhance its practical relevance and impact.
